# Eps15 Homology Domain-Containing Protein 3 Hypermethylation as a Prognostic and Predictive Marker for Colorectal Cancer

**DOI:** 10.3390/biomedicines9050453

**Published:** 2021-04-22

**Authors:** Yu-Han Wang, Shih-Ching Chang, Muhamad Ansar, Chin-Sheng Hung, Ruo-Kai Lin

**Affiliations:** 1School of Medicine, Tzu Chi University, Hualien City 97071, Taiwan; g7899nek@gmail.com; 2Division of Colon and Rectal Surgery, Department of Surgery, Taipei Veterans General Hospital, Taipei 11217, Taiwan; changsc@vghtpe.gov.tw; 3Ph.D. Program in the Clinical Drug Development of Herbal Medicine, Taipei Medical University, Taipei 11031, Taiwan; muhamadanshar919@gmail.com; 4Department of Surgery, School of Medicine, College of Medicine, Taipei Medical University, Taipei 11031, Taiwan; hungcs@tmu.edu.tw; 5Division of General Surgery, Department of Surgery, Shuang Ho Hospital, Taipei Medical University, New Taipei City 23561, Taiwan; 6Graduate Institute of Pharmacognosy, Taipei Medical University, Taipei 11031, Taiwan; 7Ph.D. Program in Drug Discovery and Development Industry, College of Pharmacy, Taipei Medical University, Taipei 11031, Taiwan; 8Master Program for Clinical Pharmacogenomics and Pharmacoproteomics, Taipei 11031, Taiwan; 9Clinical Trial Center, Taipei Medical University Hospital, Taipei 11031, Taiwan

**Keywords:** *EHD3*, colorectal cancer (CRC), DNA methylation, early detection, prognostic marker, circulating cell-free DNA (ccfDNA)

## Abstract

Colorectal cancer (CRC) arises from chromosomal instability, resulting from aberrant hypermethylation in tumor suppressor genes. This study identified hypermethylated genes in CRC and investigated how they affect clinical outcomes. Methylation levels of specific genes were analyzed from The Cancer Genome Atlas dataset and 20 breast cancer, 16 esophageal cancer, 33 lung cancer, 15 uterine cancer, 504 CRC, and 9 colon polyp tissues and 102 CRC plasma samples from a Taiwanese cohort. In the Asian cohort, *Eps15 homology domain-containing protein 3* (*EHD3*) had twofold higher methylation in 44.4% of patients with colonic polyps, 37.3% of plasma from CRC patients, and 72.6% of CRC tissues, which was connected to vascular invasion and high microsatellite instability. Furthermore, *EHD3* hypermethylation was detected in other gastrointestinal cancers. In the Asian CRC cohort, low *EHD3* mRNA expression was found in 45.1% of patients and was connected to lymph node metastasis. Multivariate Cox proportional-hazards survival analysis revealed that hypermethylation in women and low mRNA expression were associated with overall survival. In the Western CRC cohort, *EHD3* hypermethylation was also connected to overall survival and lower chemotherapy and antimetabolite response rates. In conclusion, *EHD3* hypermethylation contributes to the development of CRC in both Asian and Western populations.

## 1. Introduction

Colorectal cancer (CRC) is the fourth most prevalent cancer and the second leading cause of cancer death worldwide [1]. In Taiwan, cancer is the leading cause of death, and CRC is the third most commonly diagnosed cancer [2]. Detection of early-stage CRC is crucial for maximizing the benefits of medical intervention [3]. Multiple genetic and epigenetic modifications result in the silencing of tumor suppressor genes (TSGs) and activation of oncogenes, which transform normal colonic epithelial cells into adenocarcinoma cells in CRC [4,5]. DNA methylation is the most common and vital epigenetic mechanism [6]. In gene promoter regions, aberrant DNA methylation can inactivate TSGs, contributing to tumorigenesis [7,8,9]. DNA methylation often occurs in multiple independent CpG islands, meaning that universal DNA methylation occurs in different cancers [10,11]. A global analysis of 98 primary human tumors revealed an average of 600 CpG islands with abnormal methylation among a total of 45,000 in the genome [12]. Accordingly, the recognition of methylation patterns can explain the expression of various CRC subtypes [9]. Moreover, the development of new biomarkers and new therapeutic interventions is indispensable for the improved diagnosis and prognosis of CRC.

Analysis of circulating cell-free DNA (ccfDNA) can provide genetic and epigenetic information related to CRC [13]. With advantages of minimal invasion and cost effectiveness, ccfDNA methylation assessment might be a vital diagnosis and prognosis tool in precision oncology. For instance, detection of Septin 9 hypermethylation DNA in plasma has been approved by the US Food and Drug Administration for CRC screening. Although a cancer constantly evolves, treatment often remains based on the first tumor biopsy result. Therefore, real-time therapy response monitoring through ccfDNA analysis should be considered for personalized and precision medicine [13]. Together, these findings encouraged us to investigate novel DNA methylation biomarkers for CRC, aiming to provide information for future CRC diagnostic and treatment decisions.

The Illumina Infinium HumanMethylation450 BeadChip (450K) array is a high-resolution, genome-wide approach for detecting DNA methylation pattern abnormalities in cancer [14,15]. With this panel, we found different loci in hypermethylated promoter and exon 1 regions of the Eps15 homology domain-containing protein 3 *(EHD3)* gene after comparing 26 paired normal colorectal tissues and CRC tissues. *EHD3* includes 35,300 nucleotides and its sequence, which encodes a predicted protein of 535 amino acids, is located on chromosome 2p23.1. The alignment of *EHD3* sequences from UniProtKB (Q9NZN3) disclosed one EH domain within the *EHD3* protein sequence. The EH domain (for the Eps15 homology domain) was found in the tyrosine kinase substrate Eps15 and endocytosis, vesicle transport, and signal transduction proteins [16,17]. The *EHD3* protein regulates the endosome-to-Golgi transport and downstream lysosomal biosynthetic transport pathways [18,19]. Protein expression of *EHD3* was reported to be low in glioma [20]. Aberrant DNA methylation was previously found in esophageal squamous cell carcinoma [21]. To date, how the DNA methylation of *EHD3* affects CRC remains unknown. We assumed responsibility for discovering the exact relationship between alteration of *EHD3* and colorectal tumorigenesis.

## 2. Materials and Methods

### 2.1. Tissue Specimens

A total of 20 frozen human breast cancer tissue samples, 16 frozen human esophageal cancer tissue samples, 33 frozen human lung cancer tissue samples, 15 frozen human uterine cancer tissue samples, 504 frozen human CRC tissue samples, 102 CRC plasma samples, and 9 polyp tissues were retrieved from Taipei Veterans General Hospital Biobank and Taipei Medical University (TMU) Joint Biobank for methylation array analysis (Figure 1D). All specimens and clinical data were collected from patients who underwent surgery at TMU and Taipei Veterans General Hospital, and they provided written informed consent.

### 2.2. DNA, ccfDNA, and RNA Extraction

This study used the RNeasy Plus Mini Kit (Qiagen, Bonn, Germany; Cat. No. 74134) for mRNA isolation and the QIAamp DNA Mini Kit (Qiagen; Cat. No. 51306) to retrieve genomic DNA from individual patients’ matched pairs of tumors and adjacent normal tissue. ccfDNA was prepared from 200 μL of plasma using a MagMAX Cell-Free DNA Isolation Kit (Thermo Scientific, Austin, TX, USA; Cat. No. A29319), and cDNA was prepared using the iScript cDNA Synthesis Kit (Bio-Rad Laboratories, Shanghai, China; Cat. No. 170-8891). Finally, DNA, ccfDNA, and RNA were quantified using Thermo Scientific NanoDrop 2000 c spectrophotometers to measure the A260/A280 ratios.

### 2.3. Assessment of Genome-Wide Methylation Level

After sodium bisulfite conversion of 26 paired CRC tissues and corresponding noncancerous colon tissues by EpiTect Fast Bisulfite Conversion Kits (Qiagen, Bonn, Germany; Cat. No. 59826) was completed, we used 450K (Illumina, San Diego, CA, USA) to perform genome-wide methylation analysis. Hypermethylation or hypomethylation was detected in approximately 450,000 CpG sites in this array through the use of designed target-specific probes. The methylation level was represented as β values ranging from 0 (no methylation) to 1 (full methylation) because it was calculated as the number of methylated signal outputs divided by total outputs.

### 2.4. Real-Time Reverse-Transcription Polymerase Chain Reaction

The LightCycler 480 Probe Master kit (Roche Applied Science, Mannheim, Germany) was used to conduct real-time reverse-transcription (RT) polymerase chain reaction (PCR) to measure the mRNA expression levels. Real-time PCR with specific primers and probe was achieved and detected using the LightCycler 480 Probe Master kit (Roche Applied Science, Indianapolis, IN, USA; Cat. No. 04707494001). The reference gene was glyceraldehyde-3-phosphate dehydrogenase (GAPDH).

The mRNA expression values normalized with GAPDH were calculated using LightCycler Relative Quantification software (Version 2.0, Roche Applied Science). If the mRNA expression level relative to GAPDH in CRC tumor tissue was 1.5-fold higher than that in paired normal colorectal tissue, the mRNA expression was considered high. If the mRNA expression level relative to the control group was 0.5-fold lower, the mRNA expression was considered low. Appendix A present the primers.

### 2.5. TaqMan Quantitative Methylation-Specific PCR

The EpiTect Fast DNA Bisulfite Conversion Kit (Qiagen; Cat. No. 59826) was used in the bisulfite conversion of genomic DNA. The level of DNA methylation was measured using a TaqMan quantitative methylation-specific PCR (QMSP). Our lab performed QMSP with specific primers and a methyl-TaqMan probe using the SensiFAST^TM^ Probe No-ROX Kit (Bioline, London, UK; Cat. No. BIO-86020). The reference gene was beta-actin (ACTB). Appendix A presents the primers.

### 2.6. TCGA Data Analysis and Candidate Gene Selection

All Western cohort results, including gene expression, methylation condition, and clinical information, were based on data generated by the TCGA Research Network (available online: http://cancergenome.nih.gov/ (accessed on 27 July 2020)). Gene expression was considered high (low) when the mRNA value of the tumor tissue was ≥1.5 (≤0.5) times that of the paired adjacent normal tissues.

### 2.7. Statistical Analyses

All statistical results were conducted using SPSS (IBM Corp. Released 2017. IBM SPSS Statistics for Macintosh, Version 25.0. Armonk, NY, USA: IBM Corp.). Pearson’s chi-square test was used to characterize *EHD3* methylation, mRNA expression, and protein expression level in patients with CRC with respect to clinical data, including age, sex, tumor type, and TNM tumor stage. Calculated overall survival curves were constructed using the Kaplan–Meier method and multivariate Cox proportional-hazards survival analysis. The significance criterion for survival analysis was log-rank *p* < 0.05.

## 3. Results

### 3.1. EHD3 Is a Common Target in Alimentary Canal Cancer

To identify a new drug-designed target for gastrointestinal cancer, this study analyzed the methylation patterns of patients with esophageal, gastric, and colon cancer from TCGA. First, we excluded CpG sites with average β values of >0.05 for noncancerous tissues and <0.30 for tumor tissues (Figure 1A). Accordingly, 919 CpG sites from colon cancer, 442 from rectal cancer, and 291 from gastric cancer were identified. Second, 43 CpG sites were identified at the intersection of the 3 aforementioned categories. Finally, to determine a common gene target for both Western and Asian cohorts, 450K was used to analyze the paired noncancerous colon tissues and CRC tissues of 26 patients (Appendix A), and 8 CpG sites from 6 genes (*MSC*, *ZNF132*, *PDGFD*, *BEND5*, *EHD3*, and *SOX5*) were among the 100 most common ΔAvg_β (βTumor–βNormal) CpG sites of colon cancer in Taiwan (Figure 1B). Few reports on *EHD3* hypermethylation in cancer have been published; therefore, *EHD3* was selected for further analysis.

Cancers originating outside the alimentary canal exhibited minimal *EHD3* hypermethylation in both Western and Asian populations. From the TCGA dataset, only small portions of breast cancers (1.1%, 1/87), lung adenocarcinomas (6.9%, 2/29), and squamous cell lung cancers (0.0%, 0/40) had *EHD3* hypermethylation (Appendix A). Similar results were found in the Asian cohort. *EHD3* hypermethylation was at least twofold higher in tumors than in matched normal tissues in only 15.2% (5/33) of patients with lung cancer, 40.0% (8/20) of those with breast cancer, and 13.3% (2/15) of those with uterine cancer (Appendix A).

*EHD3* hypermethylation was observed in most Western alimentary canal cancers compared with paired adjacent normal tissues (Figure 1C), such as colon cancer (68.4%, 26/38), esophageal cancer (60.0%, 9/15), liver cancer (33.3%, 4/12), gastric cancer (2/2, 100%), pancreatic cancer (10.0%, 1/10), and rectal cancer (42.9%, 3/7). In the Taiwanese cohort, *EHD3* hypermethylation was detected in 73.1% (19/26) of patients with CRC and 18.8% (3/16) of patients with esophageal cancer compared with matched normal tissues (Figure 2A and Appendix A). In both the Western and Asian cohorts, because *EHD3* hypermethylation was only steadily performed in patients with CRC, the biological role of *EHD3* in CRC was the next target to understand in the present study.

### 3.2. EHD3 Promoter Hypermethylation and Low Expression of mRNA and Protein in Asian Patients with CRC

We used 450K to analyze 26 patients’ paired noncancerous colon tissues and CRC tissues. The promoter and exon 1 regions of *EHD3* in CRC tumor tissues exhibited 21 highly methylated sites (Figure 2A). Probes 4–16 were identified using a paired *t* test, and significantly more hypermethylation occurred in tumors than in normal tissues. Additionally, the average β values of probes 7, 9, and 12 were at least 0.4 times higher in tumor tissues. *EHD3* methylation patterns were confirmed using TaqMan QMSP assays in 504 patients with CRC. Because the promoter and exon 1 regions were the most hypermethylated areas in the *EHD3* gene of patients with CRC, primers and probes were designed between promoter and exon 1 regions (Figure 2, Appendix A). The data revealed that in 72.6% (366/504) of patients with CRC, *EHD3* hypermethylation was at least twofold higher in tumors than in matched normal tissues. In addition, 64.7% and 55.4% of patients with CRC had threefold and fivefold higher *EHD3* hypermethylation in tumor tissues, respectively (Table 1 and Appendix A).

*EHD3* hypermethylation plays a role in vascular invasion and high microsatellite instability in patients with CRC (Table 1, *p* = 0.005 and *p* = 0.047). Four out of nine patients (44.4%) with a benign tubular adenoma had *EHD3* hypermethylation (Figure 3A). A significant difference was detected in the Mann–Whitney *U* test between adjacent normal colon tissues and tubular adenomas (*p* = 0.042) and between tubular adenomas and CRC tumors (*p* = 0.017). Furthermore, 37.3% (38/102) of patients with CRC had *EHD3* promoter hypermethylation in plasma ccfDNA.

Whether *EHD3* hypermethylation affects mRNA expression remains unknown. Therefore, we analyzed *EHD3* mRNA expression in 102 paired CRC tissues. *EHD3* mRNA expression was at least 2-fold lower in tumor tissues than in normal tissues in 45.1% (46/102) of the paired tissues (Appendix A). The low expression also contributed to tumor metastasis to regional lymph nodes (Table 1, *p* = 0.016). In contrast to the normal tissues, 100 times higher hypermethylation in tumor tissues exhibited a statistically significant Spearman correlation with 20 times lower mRNA expression in colorectal tumors (Figure 3B, Spearman’s ρ = 0.307, *p* = 0.020). *EHD3* mRNA expression in CRC cell lines was also lower than that in normal tissues and breast and lung cancer cell lines (Appendix A).

Female patients with *EHD3* hypermethylation had poor prognosis results (Figure 3C, *p =* 0.046). Multivariate Cox proportional-hazards survival analysis revealed that *EHD3* hypermethylation in women was independently and significantly associated with poor overall survival and recurrence-free survival, even after adjustment for age, location, tumor differentiation, and cancer stage (Table 2, *p* = 0.020 and *p* = 0.020). Another Cox regression analysis adjusted for sex, age, location, differentiation, and stage revealed that low *EHD3* mRNA expression was significantly associated with recurrence-free survival (Table 2, *p* = 0.029).

### 3.3. Promoter Hypermethylation, Low mRNA, and Protein Expression of EHD3 in Western Patients with CRC and Poor Prognoses

We investigated whether the *EHD3* methylation pattern was different in a CRC cohort outside Asia. Probes 4–16 were identified using a paired t test and indicated significantly higher hypermethylation in tumors than in normal tissues. Additionally, the average β value of probe 12 was at least 0.4 times higher in tumor tissues.

Analysis of RNA and protein data indicated that *EHD3* RNA and protein expression was significantly reduced in CRC tumor tissues compared with matched normal tissues (*p* = 0.029 and *p* < 0.001) [23]. The Pearson correlation test revealed that for probes 1–15 in promotor and exon 1, *EHD3* mRNA expression and *EHD3* hypermethylation were significantly, and at least modestly, negatively correlated. Moreover, the test indicated a moderately negative correlation in probe 16, which was in exon 1 (Pearson correlation = −0.416, *p* < 0.001, Figure 4A, *n* = 289 tumors). By contrast, the correlation was positive in the *EHD3* gene body region and three-prime untranslated region (array probes 18–21, *p* < 0.05). Prognoses were poorer in patients exhibiting hypermethylation than in those exhibiting low methylation of *EHD3* promoter probe 4 (Figure 4B, *p* = 0.046). Multivariate Cox proportional-hazards survival analysis indicated that *EHD3* hypermethylation was independently and significantly associated with overall survival after adjustment for age, location, tumor size, lymph invasion, and metastasis (Appendix A, *p* = 0.039). Patients with late-stage cancer and female patients with late-stage and low EHD3 protein expression consistently had poorer prognoses than did those with high EHD3 protein expression (Figure 4C,D, *p* = 0.045 and *p* = 0.012) [24].

### 3.4. Promoter Hypermethylation of EHD3 Reduces Drug Sensitivity in Patients with CRC from Western Countries

To further determine *EHD3* hypermethylation in CRC from the TCGA dataset, this study analyzed the correlation of the treatment outcomes of antimetabolites, DNA alkylating drugs, and topoisomerase inhibitors with the methylation level of the *EHD3* promoter region. As a result, *EHD3* promoter (cg01163837) hypermethylation was significantly associated with poor treatment response to chemotherapy and antimetabolites (*p* = 0.017 and *p* = 0.039, Table 3).

## 4. Discussion

The silencing of TSGs could be caused by hypermethylation of CpG islands, resulting in tumorigenesis. We used 450K and discovered that the promoter and exon 1 regions of *EHD3* had multiple highly methylated CpG sites in CRC tissues but not in corresponding normal tissues. Furthermore, QMSP confirmed *EHD3* hypermethylation in CRC tissues compared with normal tissues.

In the Asian cohort, female patients with CRC and *EHD3* promoter hypermethylation had poor prognoses (Figure 3C). Cox regression analysis indicated that low *EHD3* mRNA expression in Asian patients with CRC was also associated with poor recurrence-free survival (Table 2, *p* = 0.029). Caucasian patients with CRC and high *EHD3* promoter methylation (Figure 4B) or low *EHD3* protein expression (Figure 4C) also had poor prognosis. Vascular invasion and lymphatic invasion, both poor prognostic factors for CRC [25,26], were observed in patients with CRC with high *EHD3* methylation and low *EHD3* mRNA expression (Table 1). These two factors may explain why *EHD3* was involved in clinical prognosis. A previous study suggested that *EHD3* polymorphism explains why women are more prone to major depressive disorder than men are [27]. Therefore, another explanation for correlation between *EHD3* hypermethylation and poor prognoses is that women with CRC and *EHD3* hypermethylation may present signs of mental illness, leading to a poor prognosis.

The epigenetic pattern of promoter hypermethylation and gene body demethylation in *EHD3* sequences was generally similar between the CRC tissues of the Asian and Caucasian cohorts (Figure 2). Naturally, the correlation between *EHD3* promoter hypermethylation and reduced *EHD3* transcript level was significant in both cohorts (Figure 3B and Figure 4A). Because *EHD3* promoter hypermethylation was found in 72.6% of Taiwanese CRC tissues, which was higher than *SEPT9* hypermethylation found in 60.92% of Taiwanese CRC tissues [28] and in 44.4% of patients with tubular polyp adenomas, *EHD3* hypermethylation can serve as an early indicator of CRC or as a viable auxiliary to the pathological diagnosis of malignant polyps.

*EHD3* belongs to the Eps15 homology domain family, which has a module implicated in protein–protein interaction [17]; this family is involved in endocytic transport [29]. *EHD3* influences the coordination of ciliary membrane and axoneme growth [30], the stabilization of tubular recycling endosomes to help endocytic recycling [18], and the regulation of endosome-to-Golgi transport to affect lysosome transport [18]. It also functions as a TSG and induces a G0/G1 growth arrest and apoptotic cell death in glioma cells [20]. Therefore, we speculated that the loss of EHD3 expression in colon cells occurs through the same mechanism and increases cell growth and tumorigenesis.

*EHD3* promoter hypermethylation likely plays a vital role in gastrointestinal cancers, such as esophageal and gastric cancer, in Western populations (Figure 1C). It was also identified using 450K in Chinese patients with esophageal carcinoma [21]. However, QMSP analysis of Taiwanese patients with esophageal and breast cancer revealed a lower aberrant frequency of the *EHD3* promoter methylation level. Analysis of the *EHD3* mRNA expression in gastrointestinal cancers might help elucidate the relationship between the *EHD3* anomaly and cancer development. To summarize, gastrointestinal cancers are often accompanied by *EHD3* promoter hypermethylation, especially in regions involved in food transport, including the esophagus, stomach, colon, and rectum. Unhealthy personal dietary habits may cause *EHD3* hypermethylation and should be studied further.

Adjuvant chemotherapy, including antimetabolites, alkylating agents, and topoisomerase inhibitors, is often used after CRC surgery. The chemotherapy response rate was significantly higher in patients with low *EHD3* methylation levels (*p* = 0.017, Table 3). Antimetabolites are recommended for patients without *EHD3* promoter hypermethylation (*p* = 0.039, Table 3) for postoperative chemotherapy. In our study, 37.3% (38/102) of patients with CRC exhibited *EHD3* promoter hypermethylation in plasma ccfDNA; hence, targeted therapy and immunotherapy drugs may be treatment options. Regular monitoring of *EHD3* methylation level and adjusting CRC treatment accordingly are also highly recommended.

MSI (microsatellite instability) status was associated with *EHD3* methylation level in Taiwanese patients with CRC (Table 1, *p* = 0.047). Patients with MSI-H (high microsatellite instability) CRC have better overall survival but are resistant to 5-fluorouracil-based chemotherapy [21]. Similarly, patients with *EHD3* hypermethylation have poor overall survival and response to chemotherapy. Future studies should investigate the relationship between MSI status and *EHD3* methylation levels. For instance, a prospective analysis of the influence of *EHD3* hypermethylation and MSI-H on CRC drug response with an adequate sample size may help in combining *EHD3* methylation level with MSI status to develop a strong indicator for CRC treatment. Therefore, *EHD3* can not only be developed as a prognostic marker but also as a tumor biological factor for chemotherapy response.

## 5. Conclusions

*EHD3* hypermethylation promotes the development of CRC and other gastrointestinal cancers in both Asian and Western populations and can be developed as a prognostic marker or a target for precision medicine.

## Figures and Tables

**Figure 1 biomedicines-09-00453-f001:**
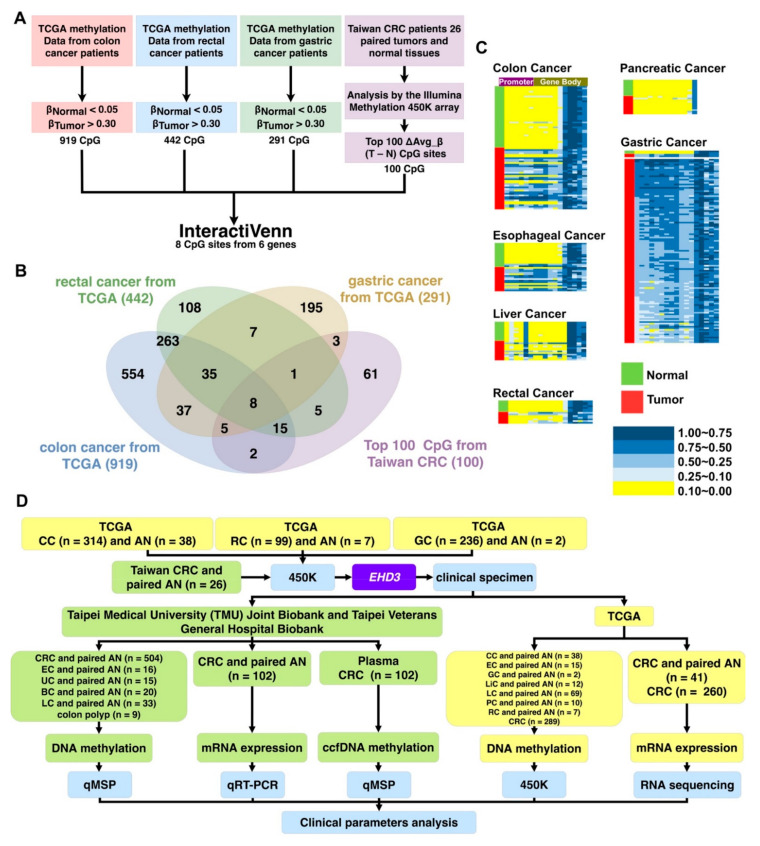
Gene selection and experimental design. (**A**) Stepwise gene selection flowchart. (**B**) Gene screening performed using http://www.interactivenn.net/ (accessed on 4 November 2020) [22]. (**C**) Methylation heatmap of *EHD3* in paired colon cancer, liver cancer, pancreatic cancer, esophageal cancer, rectal cancer, and gastric cancer tissues. (**D**) The experimental design is shown in the flowchart, and sample types and sizes are indicated. BC, breast cancer; EC, esophageal cancer; LC, lung cancer; UC, uterine cancer; CC, colon cancer; RC, rectal cancer; LiC, livesr cancer; GC, gastric cancer; PC, pancreatic cancer; CRC, colorectal cancer; AN, adjacent normal; ccfDNA, circulating cell-free DNA; QMSP, quantitative methylation-specific PCR; qRT-PCR, quantitative reverse-transcription PCR; IHC, immunohistochemistry; 450K, Illumina Infinium HumanMethylation450 BeadChip array.

**Figure 2 biomedicines-09-00453-f002:**
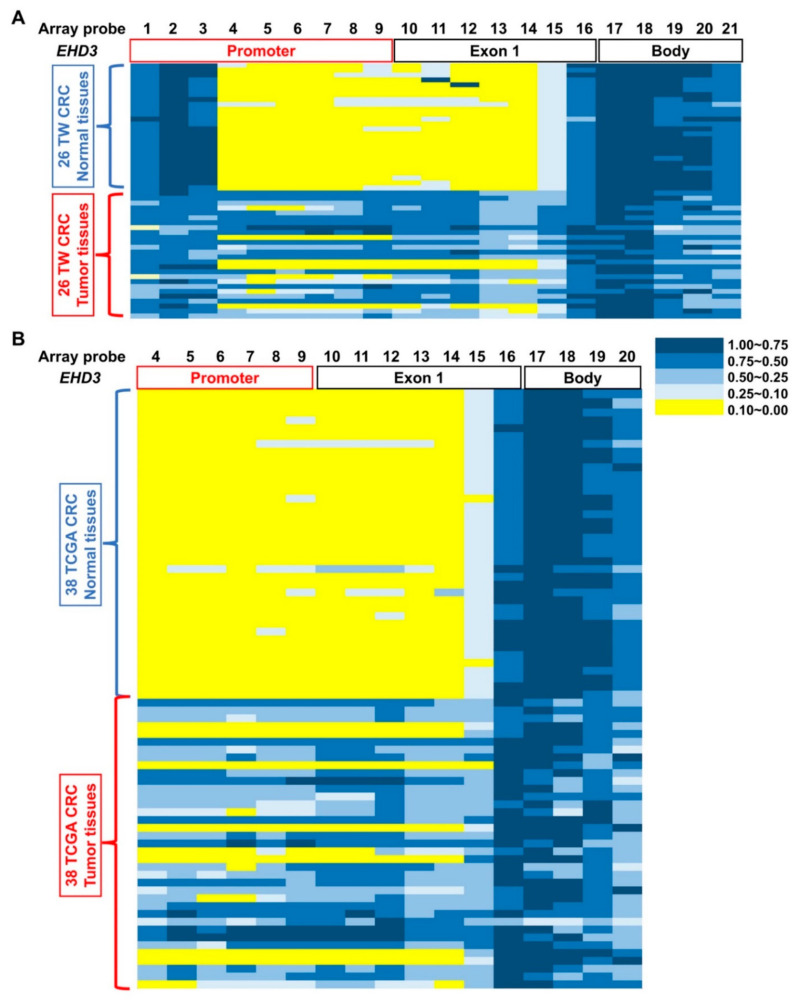
Different methylation at *EHD3* CpG islands in patients with CRC. Methylation levels (average β values) at the differentially methylated loci were identified using an Illumina Methylation 450K array-based assay in (**A**) 26 patients with CRC in Taiwan and (**B**) 38 patients with CRC from the TCGA dataset. The scale shows the relative methylation status from 0.00 to 1.00 (yellow: hypomethylation, blue: hypermethylation). Twenty-one CpG sites on *EHD3* were detected in 26 paired normal (upper) and CRC (lower) tissues, and array probes 1–21 were sites cg13149833, cg12045528, cg00648955, cg25202298, cg06773122, cg00981472, cg18444347, cg05882522, cg27230038, cg25428398, cg15355118, cg25840208, cg24743639, cg13795465, cg01163837, cg08251399, cg11957382, cg07185119, cg20203365, cg14018959, and cg14613979, respectively.

**Figure 3 biomedicines-09-00453-f003:**
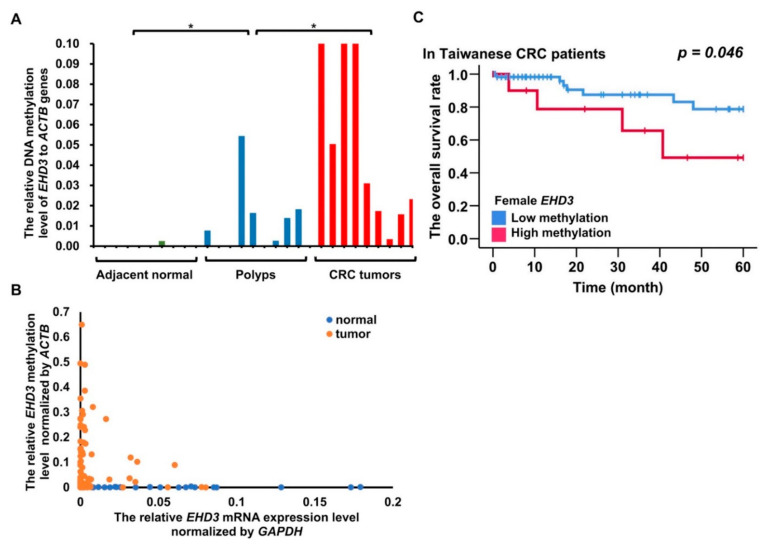
DNA methylation and mRNA expression analysis of *EHD3* from an Asian cohort. * *p* < 0.05. (**A**) *EHD3* methylation level in nine adjacent normal colon tissues, nine polyps of tubular adenoma, and nine CRC tumors. The Mann–Whitney U test was used to compare adjacent normal colon tissues and tubular adenoma and to compare tubular adenomas and CRC tumors. (**B**) Spearman’s rank-order correlation was used to estimate the correlation between *EHD3* promoter methylation and mRNA expression in the matched normal and tumor tissues. (**C**) The Kaplan–Meier estimate was used to compute the overall survival of women with and without *EHD3* hypermethylation. An *EHD3* promoter methylation level in CRC tumors 200-fold higher (90th percentile) than that in adjacent normal colorectal tissues was defined as hypermethylation.

**Figure 4 biomedicines-09-00453-f004:**
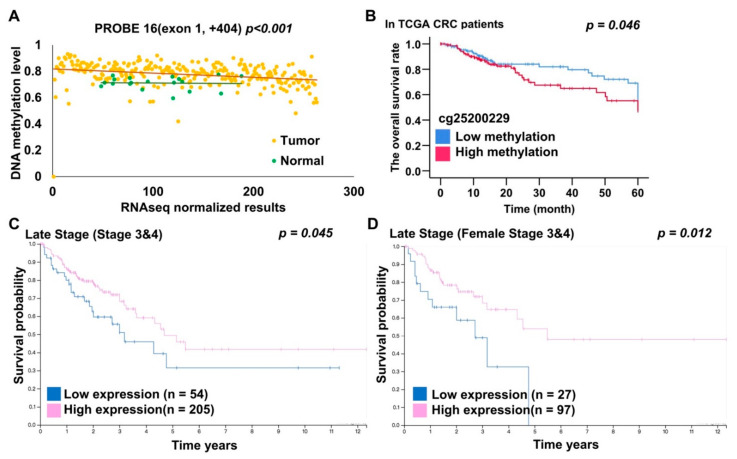
*EHD3* DNA methylation, mRNA, and protein expression analysis from TCGA dataset. (**A**) The Pearson correlation test was used to estimate the correlation between *EHD3* DNA methylation and RNA sequencing in 262 patients with CRC in the TCGA dataset. (**B**) The Kaplan–Meier estimate was used to compute the overall survival of patients with CRC with low and high *EHD3* methylation levels. *EHD3* was considered hypermethylated at an average β value of >0.445 (75th percentile). (**C**) The Kaplan–Meier estimate was used to compute the overall survival of patients with late-stage CRC with high and low EHD3 protein expression levels. Image credit: Human Protein Atlas. Image available from https://www.proteinatlas.org/ENSG00000013016-EHD3/pathology/colorectal+cancer#imid_15773055 (accessed on 27 July 2020). (**D**) The Kaplan–Meier estimate was used to compute the overall survival of female patients with late-stage CRC with high and low EHD3 protein expression levels. Image credit: Human Protein Atlas. Image available from https://www.proteinatlas.org/ENSG00000013016-EHD3/pathology/colorectal+cancer#imid_15773055 (accessed on 27 July 2020).

**Table 1 biomedicines-09-00453-t001:** *EHD3* promoter hypermethylation and mRNA expression in relation to the clinical parameters of CRC in a Taiwanese cohort ^1^.

Characteristics	Total *n* ^2^	*EHD3* Methylation ^3^				Total *n*	*EHD3* mRNA ^4^			
		Low *n* (%)		High *n* (%)			Low *n* (%)		High *n* (%)	
**CRC**	504	225	(44.6)	279	(55.4)	102	72	(70.6)	30	(29.4)
**Age**										
<65	226	104	(46.0)	122	(54.0)^0.479^	27	12	(44.4)	15	(55.6)^0.210^
>65	273	117	(42.9)	156	(57.9)	40	16	(40.0)	24	(60.0)
**Sex**										
Male	300	133	(44.3)	167	(55.7)^0.960^	39	21	(53.8)	18	(46.2)^0.960^
Female	195	86	(44.1)	109	(55.9)	28	15	(53.6)	13	(46.4)
**Tumor Type**										
Adeno	461	203	(44.0)	258	(56.0)^0.779^	65	35	(53.8)	30	(46.2)^0.439^
Others	30	14	(46.7)	16	(53.3)	1	1	(0.00)	0	(100.0)
**Tumor Stage**										
I	42	18	(42.9)	24	(57.1)^0.826^	5	3	(60.0)	2	(40.0)^0.388^
II	187	90	(48.1)	97	(51.9)	28	14	(50.0)	14	(50.0)
III	140	60	(42.9)	80	(57.1)	12	9	(75.0)	3	(25.0)
IV	96	40	(41.7)	56	(58.3)	22	10	(40.0)	12	(60.0)
**Tumor Size**										
T0-T1	34	17	(50.0)	17	(50.0)^0.494^	5	2	(40.0)	3	(60.0)^0.770^
T2-T4	455	200	(44.0)	255	(56.0)	62	29	(46.8)	33	(53.2)
**Regional lymph nodes metastasis**										
N = 0	255	120	(47.1)	135	(52.9)^0.208^	37	15	(40.5)	22	(59.5)^0.016^ *
N > 1	232	96	(41.4)	136	(58.6)	30	21	(70.0)	9	(30.0)
**Distant metastasis**										
M = 0	372	171	(46.0)	201	(54.0)^0.404^	41	23	(56.1)	18	(43.9)^0.332^
M > 1	97	40	(41.2)	57	(58.8)	23	10	(43.5)	13	(56.5)
**Differentiation grade**										
Well/Moderate	450	201	(44.7)	249	(55.3)^0.627^	63	28	(44.4)	35	(55.6)^0.876^
Poor	37	15	(40.5)	22	(59.5)	2	1	(50.0)	1	(50.0)
**Location**										
Ascending Colon	121	61	(50.4)	60	(49.6)^0.337^	15	10	(66.7)	5	(33.3)^0.242^
Transverse Colon	37	15	(40.5)	22	(59.5)	8	2	(25.0)	6	(75.0)
Descending Colon	44	15	(34.1)	29	(65.9)	7	3	(42.9)	4	(57.1)
Sigmoid Colon	152	66	(43.4)	86	(56.6)	18	7	(38.9)	11	(61.1)
Rectal	121	58	(47.9)	63	(52.1)	13	7	(53.8)	6	(46.2)
**Vascular invasion**										
No invasion	333	162	(48.6)	171	(51.4)^0.005^ **	4	2	(50.0)	2	(50.0)^0.919^
invasion	144	50	(34.7)	94	(65.3)	57	30	(52.6)	27	(47.4)
**MSI**										
MSS/MSI-L MSI-H	53	20	(37.7)	33	(62.3)^0.047^ *	32	16	(50.0)	16	(50.0)^0.677^
	7	0	(0.0)	7	(100.0)	5	2	(40.0)	3	(60.0)

* *p* < 0.05; ** *p* < 0.01. ^1^ These results were analyzed with the Pearson χ2 test. *p* values indicating significance are shown using superscripts. ^2^ For some categories, the number of samples (*n*) was less than the overall number of analyzed samples because clinical data were unavailable for them. ^3^ An *EHD3* promoter methylation level in CRC tumors fivefold higher than that in adjacent normal colorectal tissues was defined as hypermethylation. ^4^ An *EHD3* mRNA expression level in CRC tumors 0.01 times less than that of adjacent normal colorectal tissues was defined as low expression.

**Table 2 biomedicines-09-00453-t002:** Cox proportional-hazards survival analysis in Taiwanese patients with CRC.

Variable	Multivariate Analysis ^1^		
	HR	95% CI	*p*-Value
**Overall survival**			
Age	5.003	0.065–385.851	0.468
Differentiation	1.157	0.020–66.973	0.944
Tumor stage	445.234	4.525–43,808.867	0.009 ****
Location	2.538	0.271–23.731	0.414
Female *EHD3* DNA methylation ^2^	40.350	1.799–904.977	0.020 ***
*EHD3* RNA expression	0.21	0.001–0.454	0.014 ***
**Recurrence-free survival**			
Age	7.074	0.253–197.642	0.250
Differentiation	1.589	0.171–14.755	0.684
Tumor stage	174.281	5.824–5214.941	0.003 ****
Location	1.774	0.307–10.250	0.522
Female *EHD3* DNA methylation	21.966	1.630–296.011	0.020 ***
*EHD3* RNA expression	0.17	0.01–3.09	0.006 ****
**Recurrence-free survival**			
Age	1.457	0.371–5.717	0.589
Sex	0.821	0.222–3.037	0.767
Differentiation	1.355	0.234–7.833	0.734
Tumor stage	11.622	3.505–38.543	<0.000 *****
Location	1.482	0.443–4.957	0.523
*EHD3* DNA methylation	1.878	0.567–6.217	0.302
*EHD3* RNA expression	0.283	0.091–0.881	0.029 ***

* *p* < 0.05; ** *p* < 0.01; *** *p* < 0.001. ^1^ For multivariate Cox proportional-hazards survival analysis, the data were adjusted for age, differentiation, sex, tumor stage, and location. ^2^
*EHD3* methylation levels were derived from CRC tumors of 504 patients using QMSP.

**Table 3 biomedicines-09-00453-t003:** *EHD3* promoter (cg01163837) hypermethylation in relation to drug treatment response in the TCGA cohort.

Characteristics	Total	Complete Response *n* (%)	Progressive Disease *n* (%)	*p* Value ^1^
**Chemotherapy ^2^**	25	13 (52.0)	12 (48.0)	
Low methylation	13	10 (76.9)	3 (23.1)	0.017 *
High methylation	12	3 (25.0)	9 (75.0)	
**Antimetabolites**				
Low methylation	12	9 (75.0)	3 (25.0)	0.039 *
High methylation	12	3 (25.0)	9 (75.0)	
**DNA Alkylating drugs**				
Low methylation	6	5 (83.3)	1 (16.7)	0.545
High methylation	6	3 (50.0)	3 (50.0)	
**Topoisomerase inhibitors**				
Low methylation	4	2 (50.0)	2 (50.0)	0.491
High methylation	7	1 (14.3)	6 (85.7)	
**Targeted Molecular therapy ^3^**				
Low methylation	4	3 (75.0)	1 (25.0)	0.524
High methylation	6	2 (33.3)	4 (66.7)	
**Antimetabolites**				
**5-fluorouracil**				
Low methylation	8	5 (62.5)	3 (37.5)	0.074
High methylation	11	2 (18.2)	9 (81.8)	
**Capecitabine**				
Low methylation	5	5 (100.0)	0 (0.00)	0.375
High methylation	3	2 (66.7)	1(33.3)	
**Folinic acid**				
Low methylation	6	3 (50.0)	3 (50.0)	0.600
High methylation	11	3 (27.3)	8 (72.7)	

* *p* < 0.05. ^1^ These results were analyzed with Fisher’s exact test. ^2^ Antimetabolite drugs: 5-fluorouracil, capecitabine, and folinic acid; alkylating drugs: oxaliplatin and mitomycin; topoisomerase inhibitor: irinotecan. ^3^ Targeted molecular therapy: bevacizumab, regorafenib, and cetuximab.

## Data Availability

The results published here are in part based upon data generated by the TCGA Research Network: https://www.cancer.gov/tcga (accessed on 6 June 2016).

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
