# Peer review of "Eps15 Homology Domain-Containing Protein 3 Hypermethylation as a Prognostic and Predictive Marker for Colorectal Cancer"

_biomedicines, 2021, doi:10.3390/biomedicines9050453_

Round 1

Reviewer 1 Report

I read with attention the manuscript entitled " Eps15 homology domain-containing protein 3 hypermethylation as a prognostic and predictive marker for colorectal cancer ", submitted by Yu-Han Wang and co-authors. The study was aimed to identified hypermethylated genes in Colorectal cancer and investigated how they affect clinical outcomes.

In regard to a very complex work with a large amount of results, the data presentation don't always seem to follow a consequential order. The reader laboriously orients itself in reading the results due to an experimental design that is not always consistent, losing valuable information. It may be useful to include in the introduction session the experimental design to achieve the purpose of the work. Moreover, Authors should consider to join results presentation and discussion.

  • Just for example: based on abstract session, Colorectal cancer samples were indicated to be 504; but in figure 1 Colorectal cancer samples result to be more than 504 (504+260+289), please clarify this apparent Otherwise, from Table 2 multivariate analysis shown the methylataion statistically significant in female. Could Authour discuss data for male? And where not reported (fig 2, fig 3 A etc…) was the data sample made of both male and female? In the same table 2 was presented 2 times data for “Recurrence-free survival” but data from multivariate analysis for “age” and “tumour stage” are not the same (even though the statistically significant was maintained). Please specify the experimental design.

Moreover, for results reported in figure 3 (A): How the Authors chosen the subset of 9 samples? More details in experimental design and subset analysed sample are necessary.

Discussion:

  • From line 301 to line 305 I found the speculative passage to strong; the cited work speaks in favour of a gender-specific association of ehd3 polymorphisms with major depressive disorder; not for hypermethylation and mental illness. The sentence need to be supported with more scientific evidences.
  • Line 331: Authors reported: “Unhealthy personal dietary habits may cause EHD3 hypermethylation and should be studied further”. If author think it's a possible discussion and prospective they have to support their hypothesis with experimental and scientific literature evidence.
  • Why Authors focused the analysis only on female samples (figure 3C)? Despite a sample composed of 300 males (from table 1)
  • Supplementary Figure 3C represents Hypermethylation of the EHD3 promoter gene in patients with CRC, Authors can consider whether to present this graph in the main text as a general result.

Minor revision

  • Please check the letter size from line 117 to 120
  • Supplementary Figure 2: please, report in order the graphs presented and their list in the legend figure (indicating with A)…B)…etc), Moreover, how do the authors explain the ever-constant level of their EHD3 methylation in control?

Reviewer 2 Report

The article by Wang et al. is a critically designed study where the authors have comparatively analyzed a Taiwanese (Asian) cohort and an European cohort (already published in database) for various alimentary canal cancers like GC, CRC, esophageal cancer, liver cancer as well as breast, lung and uterine cancer as controls. The authors could detect significant hypermethylation in the promoter and exon 1 in a tumor suppresor gene, EHD3 using a 450K array analysis in case of CRCs and confirmed the data using Taqman chemistry. Also, they found reduced expression level of EHD3 gene as a result of this hypermethylation. They have performed extensive correlation analysis and came up with the hypotheses that this gene can be used as a risk assessment as well as prognostic marker. Moreover, this hypermethylation event could be a potential drug target to slow down CRCs. The experiments are designed well and results are supported by evidences. The manuscript is well written overall. This reviewer has some minor suggestions before this article being accepted for publication.

General comments:

  1. CRCs are categorized into two types: Microsatellite stable (MSS) and sporadic (Microsatellite unstable, MSI). Did the author had any preference for MSS or MSI among the patient cohort or did they had any prior information while collecting the patient samples? Any comment on that will be useful.
  2. I do not see any link for public deposition of the array data for Taiwanese (Asian) cohort. If the data is already deposited, then the link of the biorepository should be presented. If not, this data should be deposited in some suitable public domain for future use by the researchers. For eg., in recent time, it was reported that DNA repair gene NEIL2 is downregulated in the CRC population, however, the mechanism of such downregulation could not be assessed (Syed et al. Cells, 2020). From this cohort array analysis, is there any indication of hypermethylation of NEIL2 promoter or exon regions? These kind of questions could be easily explored with the array data, if published.
  3. For the expression level analysis of EHD3, mostly qRT-PCR has been carried out. Is it possible for the authors to check the downregulation in the protein level by representative Western blots? At least a few patient samples could be tested.
  4. It will be difficult for the common readers to follow ß-value without any proper introduction to it. It should be incorporated in the text what this actually stands for. Also, the particular parameters used for this value in choosing CpG sites (>0.05 for noncancerous tissues and <0.30 for tumor tissues) should be elaborately explained. Are these arbitrarily chosen or there is a set standard for this? It should be cited.
  5. Similarly, authors should explain the basis for choosing the gene expression parameter (1.5 fold high, 0.5 fold low). Is it chosen arbitrarily or there is a set standard for this kind of assays?

Minor comments:

Introduction: The last paragraph where the results are summarized should be written more clearly.

Methods: The tables and figures should be numbered according to the order they are cited in the text. Like Supplementary Table 3 and supplementary Fig. 3A are cited in the method.

Results: Section 3.1: “Few reports on EHD3 hypermethylation in cancer has been published”. Please cite these reports.

Fig. 1C: It is difficult to follow the color codes and corresponding parameters in the heat-map. These should be more clearly described in the corresponding figure legends.

Fig. 1D: It should be cited in the text.

Section 3.4: Needs more elaboration. A little more background information of these drugs needs to be mentioned and then in that context, the data should be explained.

Round 2

Reviewer 1 Report

The resubmitted version of the manuscript entitled "Eps15 homology domain-containing protein 3 hypermethylation as a prognostic and predictive marker for colorectal cancer", substantially includes the requested suggestions even if, I do not see the difference between the new and the old Figure 1 D, could the Authors explain me their purpose as minor revision.  In regard to a large amount of results, the current data presentation in the tables and in the text now appears more clear.